# An Entrepreneurship Incubation Process Model and Gamified Educational Software Designed for Sustainable Education

Ping Liu 

School of International Business and Management, Sichuan International Studies University, Chongqing 400031, China; liuping@sisu.edu.cn

**Abstract:** To tackle the challenges of the sustainable development of entrepreneurship education in China and bridge the gaps between academic research and practices for Goal 4 of the SDGs, this design science research aims to create an educational artifact for the incubation of entrepreneurship from students' tacit knowledge and evaluate whether it performs well. First, the incubation process of the educational artifact is summarized, including the functions and tools of its supporting software. An Extended Triple Diamond model with design principles in the knowledge creation process is proposed in this research. Then, the educational artifact is evaluated based on the data from in-depth interviews conducted with 15 university teacher and student users. Through the evaluation, functions of the supporting software and some organizational arrangements of the activity are iterated. This design science research of the educational artifact contributes empirically to the body of design knowledge. In terms of practice, the educational artifact with extremely low entry barriers is expected to alleviate the inherent contradictions between personalization (the inherent characteristics of innovation and entrepreneurship) and large-scale development (to be carried out in a sustainable manner), thus contributing to the Sustainable Development Goal of inclusive and equitable quality education.

**Keywords:** sustainable education; gamified educational software; innovation and entrepreneurship; design thinking; design science

## 1. Introduction

Entrepreneurship plays increasingly important roles in today's VUCA (volatile, uncertain, complex, and ambiguous) world. Effectually educating students on entrepreneurship and innovation remains challenging. The 2030 Agenda for Sustainable Development calls for action to change our world [1]. In China, former Premier Li Keqiang proposed the encouragement of Mass Entrepreneurship and Innovation at the opening ceremony of the Eighth Summer Davos Forum in September 2014. Since then, the Chinese government, at different levels, has issued a series of policies. Mass Entrepreneurship and Innovation quickly became a national strategy and was implemented nationwide in China. One of the latest policies issued in September 2021, in which university students are targeted as the new and vital force to achieve sustainable development goals.

However, whether intuitively or empirically speaking, innovation and entrepreneurship are difficult and are typically matters for a small number of people. The entrepreneurial population usually accounts for less than 5% of the total population in a country. In the past ten years (2010–2020), the proportion of Chinese college graduates who started their own businesses did not exceed 3% [2]. How could the innovation and entrepreneurship activities that originally only related to a small number of people be conducted on a larger scale? Achieving the sustainable development goals of innovation and entrepreneurship in China appears to be a significant challenge.

To tackle this challenge and bridge the gaps between academic research and practice, the design science approach is adopted in this research to create an artifact of entrepreneur-

ship incubation model, including process and design principles. Then the artifact is evaluated using a deep understanding of the feedback from its teacher and student users. Its supporting gamified educational software, named Fire Festival, is also provided, through which personalized entrepreneurship education activities can be conducted on a large scale with an extremely low cost of participation.

By doing so, it significantly lowers the barrier to Chinese college students' participation in entrepreneurship activities and, therefore, contributes to the fourth Sustainable Development Goal (SDG) initiated by the United Nations to "ensure inclusive and equitable quality education and promote lifelong learning opportunities for all" [1]. It also contributes to the outcome targets of SDG 4 that ensure equal access for all to quality and affordable higher education by 2030 [3].

To deal with the sustainable education challenges identified above, an educational artifact is designed in this design science research. The objectives of the study are as follows:

(1) Create a process model along with supporting software and provide guiding principles for teachers to facilitate the process of knowledge conversion;
(2) Evaluate its performance based on user feedback to improve and iterate the educational artifact.

## 2. Literature Review

In this section, the science of the artificial is introduced to illustrate how the ontology and epistemology of artificial phenomena are distinct from natural phenomena. The design science methodology is suitable for dealing with artificial phenomena. Then, the design science research in the field of entrepreneurship and gamification in education is reviewed. Following this, the design thinking and related theories are compared, working as the foundation of the model abstracted by this research. Finally, the SECI model, the analysis framework for the evaluation part of this research, is elaborated.

### 2.1. Science of the Artificial and Science of Design

Simon [4] distinguished between natural and artificial phenomena in his seminal book, *The Science of the Artificial*. In his book, the term "natural phenomena" refers to something that already exists and is defined by necessity, where scholars or scientists try to describe, explain, and construct models to understand or predict the "truth". Conversely, artificial phenomena are defined by contingency. In the field of the artificial, the key question is whether an artifact works or not, rather than whether it is true [5].

Simon further identified two properties in the science of the artificial [4], namely, human intentionality and environmental contingency, and suggested that in the realm of the artificial, there is no unique optimal solution. The generation of artifacts is the result of the continual interaction between the two. Furthermore, different interaction conditions may lead to different results, which makes the natural scientific approach inadequate [6]. Therefore, the natural scientific approach is not suitable for addressing the problems of artificial phenomena, which calls for a new research paradigm to deal with systems that do not yet exist and need to be created to find solutions.

Therefore, in the field of the artificial, design is regarded as a third mode of research distinct from science and humanities [5,7]. Similarly, Cross [8] proposed a "three cultures" view of knowing, namely, sciences, humanities, and design. In the culture of the sciences, the phenomenon of study is the natural world and the values of this culture are seeking truth and objectivity through experimentation, analysis, etc. In the culture of humanities, the phenomenon of study is human experience and it concerns and seeks justice through its appropriate methods. However, in the culture of design, the "third area" of education, the phenomenon of study is the artificial world, and the values of this culture are appropriateness, practicality, ingenuity, and empathy. It suggests that the ontology [4], epistemology [8,9], and methodology [5,7,10,11] of the artificial phenomena should be distinct from those of natural phenomena.



Design science research focuses on the development and performance of artifacts with the explicit intention of improving the functional performance of the artifact, and is widely used in many fields of applied science including architecture [12], engineering [13], information systems [14–18], education [19], and management [20–22]. In this research, the design science approach rather than explanatory science is adopted to create and evaluate an educational artifact.

### 2.2. Design Science in Entrepreneurship

In the field of entrepreneurship, researchers have taken design science as an important perspective to develop research with practice: focusing on entrepreneurship as a design process through the development of design principles [23–25], regarding opportunities as artifacts in entrepreneurship [5,26,27], creating artifacts to solve business problems [28,29] such as Osterwalder's business model canvas [30,31], etc. One of the most representative and groundbreaking studies of design science research in entrepreneurship is Sarasvathy's Effectuation Theory [32], focusing on problem-solving instead of finding truth or explaining causal relationships. She further develops five principles [33] to elaborate on the differences between effectuation and causation.

Dimov [5] discusses design science as a distinct mode of research and outlines a design science perspective of entrepreneurship. In addition, Dimov [27] compares different research methodologies with the design science of entrepreneurship. The descriptive science or behavioral science of entrepreneurship tries to describe the true world and explain the causal relationships of a phenomenon, while the design science of entrepreneurship aims to create instrumental and practical knowledge or provide suggestions for how the world ought to be to achieve certain goals.

Following this vein of design science in entrepreneurship, practical knowledge is provided in this research to tackle the "wicked problem" [34] of Mass Innovation and Entrepreneurship in China. The incubation artifact with process and design principles offers instrumental knowledge between means and ends, contributing to entrepreneurship teaching in higher education.

### 2.3. Gamification in Education

The gaming environment plays an increasingly important role in various fields, especially in the context of the rapid development of the Metaverse [35,36]. In terms of education, researchers treat design as a means and a pedagogy to develop education theories [37–39], take learning and teaching process as a design science [40–42], and create artifacts for education purposes [43,44]. Within this pedagogy of design science, gamification becomes an increasingly important method in education [45], in E-learning [46], and in entrepreneurship education [47–50].

The term "gamification" refers to applying game mechanisms or game design elements in non-gaming environments [45] to enhance the user experience and achieve the goal in that field effectively. According to the review study by Nah et al. [51], the game design elements could include awarding points/badges, levels/stages, leaderboards, prizes, showing progress bars, providing a storyline, and giving feedback. In this research, game mechanisms and game design elements can be found in the educational artifact and were evaluated by university teacher and student users.

### 2.4. Theory of Design Thinking

Design thinking is one of the best and most popular tools to scaffold people to innovate and is considered important in teaching entrepreneurship [52,53]. It can be found in different forms, as outlined below.

- Design Thinking: the Five-Stage Model

In 1991, David Kelley founded IDEO, a world-renowned design company, with design thinking as its core idea, and implemented it in IDEO's work. According to IDEO, there are three areas in their design thinking model: Inspiration, Ideation, and Implementation [54].

The five-stage design thinking model proposed by the Hasso Plattner Institute of Design, Stanford University (the d.school) encompasses the stages of Empathize, Define, Ideate, Prototype, and Test [55], through which students or participants can derive innovative outputs. Both models emphasize that design thinking is a non-linear process. It does not even have to follow any particular order, either in parallel or iteratively.

The theory of design thinking is one of the most advanced and popular methods to develop a product or service based on the true needs of users. The design thinking process is widely used in practice and taught in universities. For example, besides being taught in design schools, design thinking is integrated into major university courses to cultivate students' creativity, empathy, innovation ability, and entrepreneurship awareness [56–58]. Many models and artifacts are derived from design thinking theory with certain differences in steps, such as the Google Design Sprint discussed later. Design thinking is also the fundamental theory guiding the design of this research's educational artifact.

- Double Diamond: A Design Process Model

The Double Diamond is a design process model popularized by the British Design Council in 2005. It was designed with four phases: *Discover*, *Define*, *Develop*, and *Deliver* [59]. The four phases of the Double Diamond model were adapted from the divergence–convergence model proposed by Banathy [60] and included the five stages of design thinking in the two divergence–convergence processes, as shown in Figure 1.

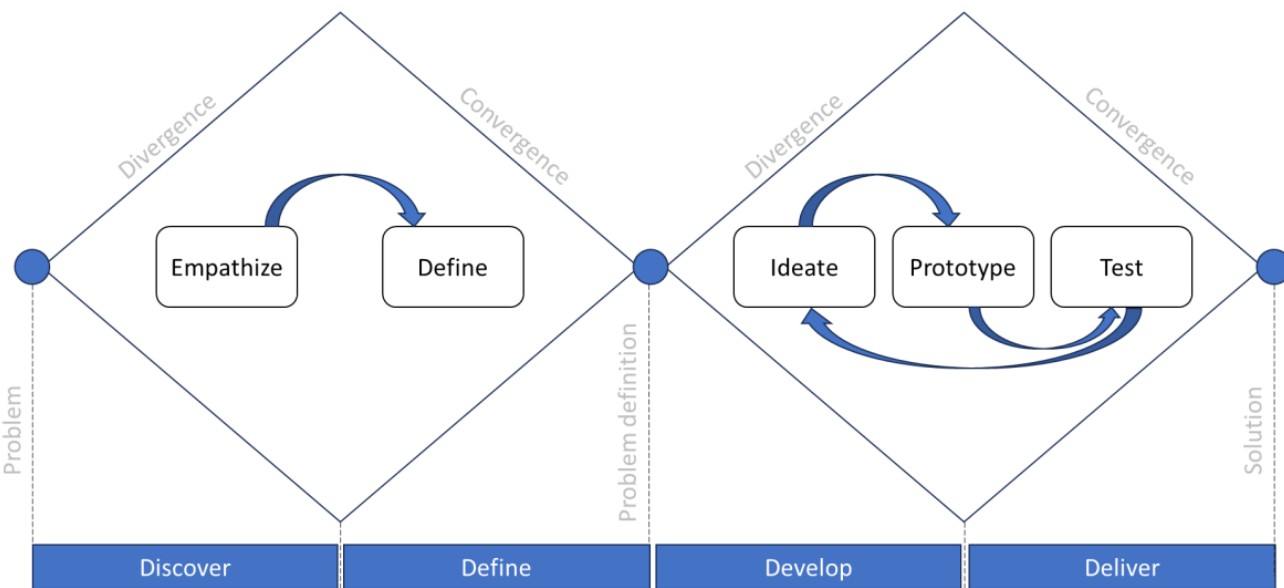

**Figure 1.** Double Diamond model of design thinking (based on the Double Diamond by the Design Council, CC BY 4.0, and the five stages of the design thinking process by the Interaction Design Foundation, CC BY-NC-SA 3.0).

In the first diamond, the problem is defined by empathizing with customers' pain points and deciding which one is the problem that the team wants to solve. After the problem definition, the task of the second diamond is to find the solution to the problem, which also goes through a process of divergence and convergence. First, as many solutions as possible are ideated in the divergence phase, and then one of these is selected for development into a prototype and is tested in the convergence phase to obtain feedback.

The biggest contribution of the Double Diamond model is the introduction of the divergence–convergence model into the design thinking process, which is beneficial to practice. It is widely used under the promotion of the British Design Council. Following this vein, a triple diamond model is proposed in this research, in which the "ideate" stage is listed as a separate divergence–convergence process to devise solutions for the defined problems.

- Design Sprint: Google Ventures Design Model

In practice, to help startups with product and business innovation, Google Ventures developed a design process for the development of new features or products named Design Sprint. Employees and participants must follow six steps—Understand, Define, Diverge, Decide, Prototype, and Validate—over five days [61], which is an adaptation of the Double Diamond model and the five-stage model of design thinking.

To be specific, compared with the five-stage model, this approach extends the step of Ideate in the design thinking process into two separate steps: Diverge and Decide. Compared with the Double Diamond, the Google Design Sprint model splits the second diamond into two diamonds, making a "triple diamond" model. In this way, all six steps become three divergence and convergence processes, which provides good rhythm for both participants and the organizers of the design activities over five days.

Compared with the six steps involved in Design Sprint, the incubation process of the current research is extended to include nine stages by adding inputs and outputs to the three times of divergence and convergence. The nine stages are elaborated in Section 4.1.1.

### 2.5. Theory of Organizational Knowledge Creation

As Polanyi [62] stated, "we can know more than we can tell". This statement points out that individual minds have incalculable tacit knowledge that can be potentially transformed into explicit knowledge. In this vein, in Nonaka's opinion, tacit knowledge is, by definition, a distinctly personal concept and can be transformed into explicit knowledge, in which the transformation is bidirectional. As Nonaka [63] asserts, "organizational knowledge is created through a continuous dialogue between tacit and explicit knowledge". He proposes four modes of knowledge creation between individuals, known as SECI.

The first mode is Socialization, which describes the process of the conversion of tacit knowledge between individuals, sometimes without language. For instance, the master passes the craftsmanship to the apprentice through demonstration, and the apprentice learns this tacit knowledge through observation, imitation, deliberate practice, etc. In this mode, shared experiences between individuals are emphasized, through which the tacit knowledge of one person can be transferred to another without the assistance of explicit methods. The second mode, Externalization, is a generalization of the conversion from tacit knowledge to explicit knowledge. The role of metaphor is underlined to make the Externalization process happen. On the contrary, the mode of Internalization involves the conversion of explicit knowledge into tacit knowledge. This mode relates to the concept of learning or assimilation we are familiar with. Action and practice are important in this mode. The last mode is Combination, which is the conversion of explicit knowledge between individuals. Individuals in organizations can create new knowledge by exchanging and combining the explicit knowledge held by each other.

It should be noted that the abbreviation of SECI does not reflect the sequence of conversion between tacit and explicit knowledge. Instead, organizational knowledge is created through continuous dialogues between these modes, and Nonaka summarizes it in a spiral way [63]. Knowledge is created between tacit knowledge and explicit knowledge through the four modes of SECI within and across organizations.

The theory of organizational knowledge creation guides the creation of design principles and the evaluation of the educational artifact in this study: (1) the artifact in this study is designed to encourage continuous dialogue between the tacit and explicit knowledge of participant students; (2) the design principles during the knowledge creation process are also summarized; and (3) the artifact designed in this study is evaluated through the lens of the four modes of knowledge conversion, SECI.

## 3. Methodology

The objective of this research is to provide solutions to the problems associated with sustainable education, and is more concerned with whether the solution works or not rather than whether the model is true or considering the casual relationship between variables.

Therefore, the design science methodology is adopted in this research to design an artifact for entrepreneurship education practice.

Other possible methodologies include qualitative research using grounded theory, which aims to construct a theoretical model to explain the influencing mechanism of different variables on entrepreneurship education, which is not the focus of this research. Action research is another possible methodology due to its iterating process. However, in action research, the participants are usually targeted as the objects to be iterated, for instance, in the research of an educational method's performance to improve students' abilities. The iterating target in this paper is the educational artifact rather than the teacher and student users.

### 3.1. Design Science Methodology

Design science research focuses on developing and improving the performance of an artifact. The researcher first produce an incubation model (which is an artifact with design principles) for use in entrepreneurship education, and then an evaluation of the artifact is conducted to improve its performance.

The structure of creating–evaluating in this design science research follows the framework proposed by Romme and Reymen [6]. These authors provide an inclusive research framework and suggest that the output of entrepreneurship research from the perspective of design science provides not only theoretical constructs and models built on them, but also values, principles, and practices. There are four research activities, namely, creating, evaluating, theorizing, and justifying. Among them, the two research activities of creating and evaluating are collectively referred to as "design", focusing on relevance in the practical world, and theorizing and justifying are collectively referred to as "validation", focusing on the rigor of the theory.

Since this study aims to provide instrumental knowledge of entrepreneurship incubation for higher education, deeply aligning with the ontology and epistemology of the science of the artificial, the research activities mainly concern the creating and evaluating aspects of Romme and Reymen's framework [6]. The outputs of this research are not only theoretical constructs and models, but also the values, principles, and practices related to the entrepreneurship education artifact.

### 3.2. Research Design

Aligning with two important research activities of the design science methodology, there are two focuses in this research: one is to create the artifact and the other is to evaluate it, as shown in Figure 2. Creating the artifact involves working with the founder of Campus VC in the EECN (Enterprise Educators China Network) to create and iterate the artifact. The artifact includes an educational process designed for incubating entrepreneurship from tacit knowledge, its supporting software, and a theoretical model abstracted from the activities. At the same time, the process of evaluation involves semi-structured in-depth interviews with university teacher and student users to obtain their feedback to further improve the artifact's performance.

The in-depth interviews with teachers and students were conducted using online video conferencing software and were recorded with the consent of the interviewees. The interview questions concerned their experiences, evaluations, and suggestions about the artifact. The in-depth interviews with the teachers and students were semi-structured, following the main line of questioning while remaining open to any new information that emerged during the interview.

The qualitative interview data were analyzed according to the rules of grounded theory [64]. Since our purpose here was to thematize their feedback rather than construct a theoretical model, we drew on grounded theory's clear presentation of the data and only used the first coding step, open coding [65], to converge the interview data into themes. All of the interviews were conducted in Chinese and the recorded data were transcribed into

Chinese test first for analysis, and then the results were translated into English for further interpretation.

**Figure 2.** Two focuses of this design science research: creating and evaluating.

### 3.3. Creating: Case Study

According to the design science research framework, there are two focuses in this research, namely, creating and evaluating. In the creating part, this research first summarized the process of the Fire Festival activities and introduced the functions of its online software and usage of the tools based on the case study of Campus VC. Then, we abstracted the incubation model, named the Extended Triple Diamond model, based on the practices and design-thinking-related theories. This section describes the case selection and data collection. A summarized process and the abstracted model are presented in Section 4.1.

#### 3.3.1. Case Selection

Campus VC is a social enterprise in China aiming to promote innovation and entrepreneurship among university students. Its vision is to become a future entrepreneurship platform or digital startup cloud and its mission is to encourage one million university students to start their own businesses within ten years. It was founded in 2015, the same year that the initiative of Mass Entrepreneurship and Innovation was put forward.

To fulfill their mission, Campus VC launched an entrepreneurial event named Fire Festival in 2016 and provided supporting software for students to use for free. The number of users of the software is listed in Table 1.

**Table 1.** Number of users of the Fire Festival software.

| As of | Student Users | Teacher Users | Projects Created |
|---|---|---|---|
| 12:00 p.m. 29 August 2023 | 85,969 | 1005 | 14,691 |

The reason why Campus VC was chosen as the case in this study rather than any university in China is that Campus VC, working in the role of industry in the Quadruple Helix university–industry–government–public model [66], is more independent and interactive than a single university due to its market-driven rather than policy-driven enterprise attributes. It actively connects with over 100 universities and colleges in China. The Fire Festival software they provide is used by 80,000 students and 1000 teachers at more than 100 colleges and universities. Rarely has the work of any single university been widely used by so many colleges and universities across China.

Therefore, Campus VC was targeted as the case study in this research. The incubation process summarized based on the practices of Fire Festival and the Extended Triple Diamond model abstracted by the researcher would be of further benefit for the sustainable development of entrepreneurship education.

### 3.3.2. Data Collection

The data mainly come from multiple in-depth interviews conducted with the founder, the company's internal archives that are open to the researcher, the open speeches from the founder regarding their key businesses, and some public material such as the company website and content published on the WeChat official account.

### *3.4. Evaluating: Empirical Data Collection*

There were two types of interviewees in the empirical study: (1) the university teachers who held the Fire Festival League competition in their universities, and (2) the university students in China who had participated in the Fire Festival and won prizes in the Fire Festival League competition.

The interviews with teachers were conducted one-to-one, while the interviews with students were conducted in focus groups. In the Fire Festival, students have to assemble a team to complete project challenges, so each of them may only be aware of part of the whole picture through the lens of their specific role in the team. Moreover, in focus group interviews, the answers between the team members can also form mutual inspiration, which is conducive to acquiring more detailed information. The in-depth semi-structured interview guide for teachers and students is shown in Appendix A.

### 3.4.1. Data Collection Process

This section describes the process of how to prepare a candidate list and conduct interviews with university teachers and students.

(1)    Preparing the candidate list for interviews

The first step is to create a targeted list of teachers and students to interview. The interviewees were targeted based on who had participated and won awards in the Fire Festival League to ensure their proficiency and familiarity with the artifact.

As of the time of data collection, the national Fire Festival League competition had been held twice. In the first competition, a total of fifteen projects from seven universities won awards, while eighteen projects from eight universities won awards in the second league. In combining the two competitions to obtain a targeted list for interviews, a total of twelve universities were identified, as shown in Table 2.

(2)    Sending out interview invitations

Interview invitations, including a participant information sheet and consent form, were sent by email to the candidate teachers and students. Five teachers and three groups of students accepted the interview invitation, as listed in Table 2. For the student interviewees, invitations were sent to team leaders who needed to further contact and convince their team members to participate in the interview. Therefore, the response rate of the student teams (9%) was lower than that of the teachers (42%). The significant difference in response rates is also due to the fact that the denominator of the teacher response rate is the number of universities, twelve, while the denominator of the student response rate is the total number of projects in the two competitions, thirty-three.

(3)    Conducting the interviews

The interviews were conducted using online meeting software at an agreed time. There was no time limit for the interviews set in advance, as the goal was to accomplish the semi-structured interview outline or achieve the interview expectations. But generally speaking, each teacher interview lasted about 1 h. The focus group interviews with students lasted about 1.5 h to allow each member to answer adequately.

(4)    Proofreading the interview minutes

The interview manuscript was proofread twice by the researcher. For each interview manuscript, the first readthrough was for proofreading and the second for fine-tuning and

double-checking. Then, the names of the participants were removed and replaced by their IDs. The qualitative data were coded through NVivo in the next stage of the research.

**Table 2.** List of colleges and universities that won awards in the league competition.

| # | University | No. of Awards in the 1st League | No. of Awards in the 2nd League | No. of Teachers Who Accepted Interviews | No. of Student Teams Who Accepted Interviews |
|---|---|---|---|---|---|
| 1 | Hebei University | 1 | | | |
| 2 | Hebei Normal University | 2 | 4 | | |
| 3 | Jiamusi Vocational College | 1 | 3 | 1 | 1 |
| 4 | Wuhan Polytechnic | 3 | | | 1 |
| 5 | Changchun University of Traditional Chinese Medicine | 3 | | | |
| 6 | Zhengzhou University of Aeronautics | 1 | 3 | 1 | 1 |
| 7 | Ocean University of China | 4 | | 1 | |
| 8 | Baotou Teachers' College | | 1 | | |
| 9 | Changzhou Vocational Institute of Engineering | | 2 | 1 | |
| 10 | Shanxi Institute of Technology | | 1 | | |
| 11 | Tianjin Foreign Studies University | | 2 | | |
| 12 | Southwest Minzu University | | 2 | 1 | |
| | Total | 15 | 18 | 5 | 3 |

### 3.4.2. Description of the Qualitative Data

All of the interviews were conducted in Chinese because this was the native language of the respondents. The data analysis was also carried out using the Chinese minutes, and the conclusions were translated into English after analysis. This ensured that important information was not lost in the data processing stage due to translation before the conclusions were drawn. The translation was conducted by the researcher who has adequate knowledge about the interview data with the assistance of translation software to avoid errors during translation. Another bilingual researcher also independently double-checked the translation to avoid introducing bias or misinterpretation of the data.

A summary of the qualitative data from the interviews in this study is shown in Table 3. All of the participants were given an identification number. The initial of the ID represents the type of interviewees: T for teachers and S for students. For the teachers, the numbers after the initial letter represent the personnel number. For the students, the two digits in the middle represent the group number, and the following A/B/C/D represents the personnel number in the group. Each interviewee was given a unique ID to facilitate citation in the study while maintaining the anonymity of the interviewees. The word count is also listed as a reference to the informativeness of each interview. The data analysis and results are presented in Section 4.2.1.

**Table 3.** Summary of the interview data.

| # | ID | Interview Methods | Interview Channel | Duration | Word Count |
|---|---|---|---|---|---|
| 1 | T01 | One-to-one in-depth interview | Online video conferencing software | 17:00–18:45 | 24 k |
| 2 | T02 | One-to-one in-depth interview | Online video conferencing software | 19:30–20:30 | 14 k |
| 3 | T03 | One-to-one in-depth interview | Online video conferencing software | 10:00–11:00 | 18 k |
| 4 | T04 | One-to-one in-depth interview | Online video conferencing software | 20:30–21:30 | 17 k |
| 5 | T05 | One-to-one in-depth interview | Online video conferencing software | 20:00–21:00 | 18 k |
| 6 | S01A/S01B/S01C | Focus group discussion | Online video conferencing software | 21:00–22:00 | 16 k |
| 7 | S02A/S02B/S02C | Focus group discussion | Online video conferencing software | 14:00–15:00 | 19 k |
| 8 | S03A/S03B/S03C/S03D | Focus group discussion | Online video conferencing software | 10:00–11:30 | 25 k |

### 3.5. Validity of the Research

For the creating part, regarding the massive internal archive data, the researcher gained insight and knowledge on the internal archive, knew where to look for suitable information, and was able to identify and extract valuable content due to working with the founder on several key businesses for more than two years. In addition, the researcher had a weekly online meeting with the founder. If there were any questions or doubts, the researcher could ask for clarification at any time.

For the evaluating part, to ensure the validity of the qualitative data in this research, only university teachers with more than three years of experience in innovation and entrepreneurship education were invited to the interviews. Table 4 lists the teachers' years of working experience related to innovation and entrepreneurship education. The invited teachers must have personally organized Fire Festival activities to ensure that they had a sufficient and in-depth understanding of the activities.

**Table 4.** Working experience of teacher interviewees.

| Interviewee ID | Department and Position | Years in Current Position | Years in Entrepreneurship Education |
|---|---|---|---|
| T01 | Teacher of Innovation, Entrepreneurship and Employment Guidance Department | 3 | 5 |
| T02 | Dean of the School of Innovation and Entrepreneurship | 3 | 12 |
| T03 | Section chief of Teaching and Research Department of Innovation and Entrepreneurship Center | 2 | 4 |
| T04 | Section chief of the Teaching and Research Section of the School of Innovation and Entrepreneurship | 1 | 10 |
| T05 | Director of Innovative Education Practice Center (Professor) | 5 | 16 |

Second, the interviewed students not only participated in the Fire Festival, but also won awards in the league competition, which ensured that they had an adequate understanding of and familiarity with the artifact. The members of each student focus group were from the same university or college to ensure a common basis for discussion, but had different roles in the Fire Festival project in order to obtain feedback from different perspectives.

Cognitive biases may be present in the feedback from the teachers and students. They may have had a tendency to overestimate the Fire Festival since they were rewarded by the competition. In order to mitigate the bias, the researcher always attempted to obtain more information regarding any potential negative evaluations when the interviewees gave positive feedback. Furthermore, confirmation biases may also have occurred in the data analysis process, as researchers can easily identify information that supports their initial notion. To mitigate this, NVivo -12 Plus was used to code the interview data line by line, and the codes were double-checked.

## 4. Data Analysis and Results

The process of Fire Festival activities and its supporting software are summarized in Section 4.1, and the Extended Triple Diamond model created in this study is also given based on design thinking theory and organizational knowledge creation theory. In Section 4.2, the feedback from the teachers and students is used in the evaluation of the artifact.

*4.1. Creating: Extended Triple Diamond Model with Design Principles*

The process of the educational artifact is summarized, different teaching modes are provided for teachers, and the Extended Triple Diamond model, refined based on the theory of design thinking and the theory of organizational knowledge creation, is given.

4.1.1. Fire Festival Process and Software

Fire Festival is one of the digital tools developed by Campus VC that aims to help university students incubate entrepreneurship from just an idea. Its slogan is "Kick off Your Entrepreneurship in Three Hours!". Students participating in the activity need to accomplish nine stages in three hours, assisted by specially developed online digital tools, in order to rapidly experience entrepreneurship. The digital tools of the Fire Festival both for PC and mobile terminals are free to use for student users to create projects, while teachers who want to create activities must apply for permission to use.

The Fire Festival activities involve nine stages in total, as shown in Figure 3, with six core stages in the middle, also called three times of divergence and convergence processes: Understand and Define demands, Diverge and Decide solutions, and Produce and Validate your prototypes. Besides the core activities, students participating in the Fire Festival process need to Team up on-site at the beginning of the activity and give a Roadshow speech to receive votes to determine winning projects. In the last stage, every team needs to discuss and carry out a Reflection on the whole process, which is crucial in order to internalize what they have learned.

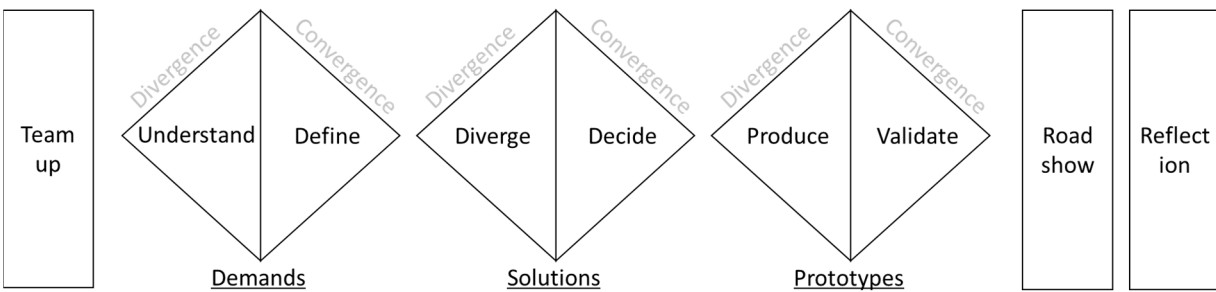

**Figure 3.** Nine stages of the Fire Festival.

(1)     Stage 1: team up

There are four tasks in the team-up stage. The first task is called One-Minute Pitch, during which every participant proposes a crazy and stupid idea. Then, each person is allowed to give a short speech, within one minute, to explain the project and the key value of it. Then, a collective vote is held to select the ideas they would like to develop further in the next stages.

Once the ideas are selected, the second task is to form a team for every proposed idea. The proposer of the idea naturally becomes the CEO of the project. Then, these CEOs are allowed to give a one-minute speech to recruit team members for their projects. Recruitment allows for two-way selection between the CEO and team members. After the team-up is completed, the participants sit together in groups.

The third task is to prepare the personnel. Each team assigns different roles and responsibilities to its members and decides on the name and mission of the team. Then, a group photo is taken with a team pose, which is uploaded the platform later.

The final task in this stage is to open the Fire Festival Mini Program (The Mini Program is a function on the WeChat application that works as a mini software program that empowers developers and businesses to launch their services on WeChat with convenient development and fluent user experience without installing apps.) on WeChat to create the project on the platform. A screenshot of the interface of the Mini Program is shown in Figure 4. The CEO creates a project by clicking the Create a Project button and fills in the information about the project; those marked with an asterisk are required. After the project

has been created, the CEO adds it to the event created by the facilitator by scanning the event QR code. Then, the team members scan the QR code to join the project created by the CEO.

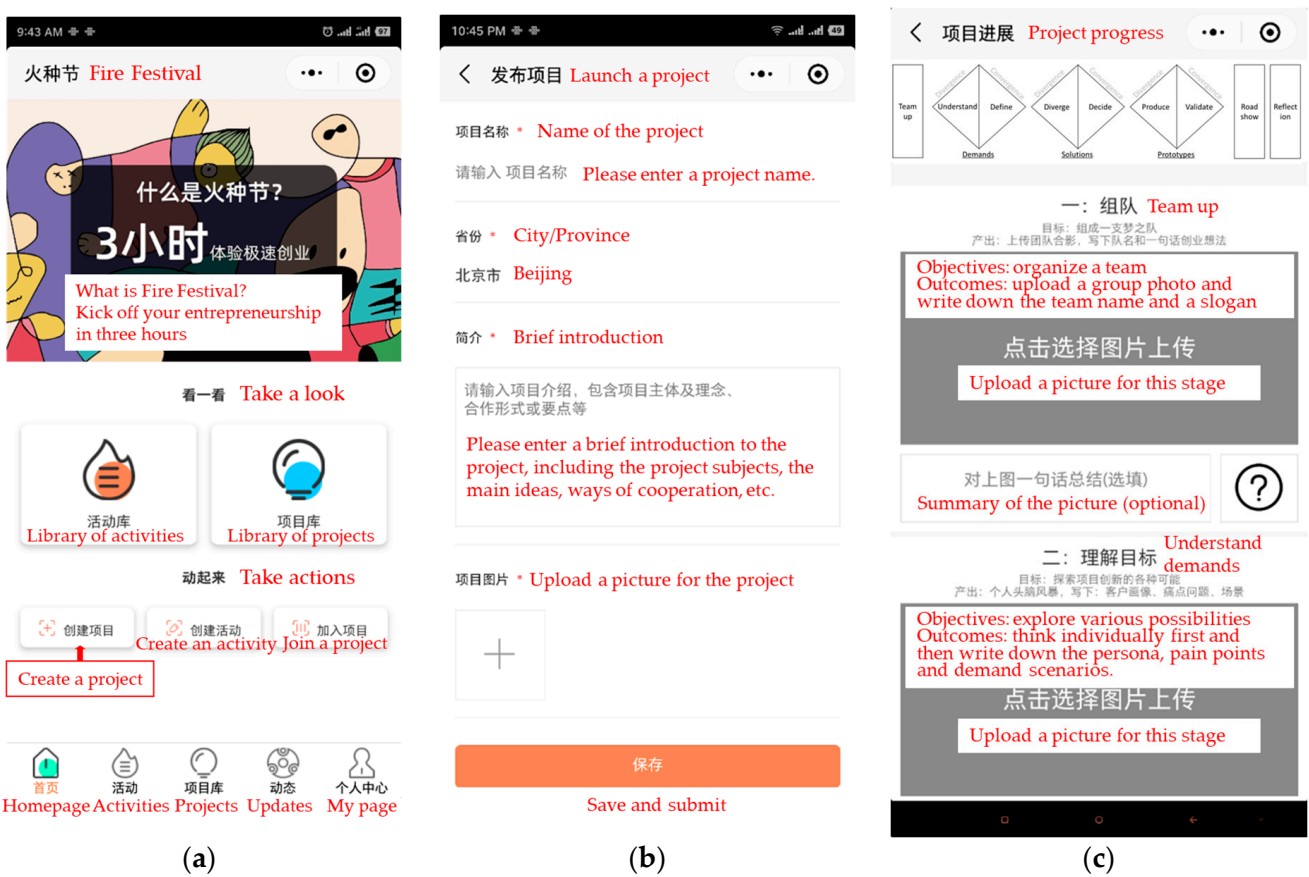

**Figure 4.** Screenshot of the interface of the Fire Festival Mini Program: (**a**) home page with option to create a project by clicking the button; (**b**) project creation page; (**c**) page showing nine stages (partial) before filling.

The nine-stage icons are gray when the project has just been created. The upper icon of each stage will automatically be lit up in color after the students complete the task. This is an incentive mechanism that motivates students to complete each stage like a checkpoint game. In addition to the color, they are also rewarded with 10 points for completing each stage. If they become stuck, the participants can click the question mark in the lower right corner of each stage to see some examples and be given hints.

(2)     Stage 2: understand the demands

After uploading the team photo and lighting up the icon of the first stage, the team can progress to the second stage: understanding customers' demands. The goal of this stage is to explore various customer needs. A piece of A4 paper is lain horizontally and an upside-down T is drawn on it to divide the paper into three areas. Then, the customer profile is written in the upper left section, the pain points on the upper right section, and the demand scenarios in the lower section of the A4 paper.

Each member of the team is required to complete the task individually first, and then a group discussion is allowed. This is the working principle running through the entire Fire Festival activity called "working independently together". For efficient cooperation within the team, direct discussion of the task with other team members without independent thinking is discouraged in the Fire Festival.

Since this stage is divergent, each member is encouraged to write down as many possibilities as they can. The customer profile refers to a group of people with common

characteristics and similar pain points. The characteristics of the target users, such as age, gender, occupation, marital status, education level, and income level, are written down. "Pain points" refers to the problems that customers are afraid of, unhappy with, and willing to spend money to solve. "Demand scenarios" refers to the time, space, and situational interactions where customers' pain points are the most intense.

When the individual part is completed, the teams present their respective answers, gather a group answer, and upload it to the platform. Then, the icon of this stage is lit up in color and 10 points are added to the project automatically.

(3)   Stage 3: define the demands

The goal of this stage is to converge and identify a specific customer's need that the team thinks is the most important. Based on the independent thinking in the previous stage, the teams conduct group discussions to clarify the target customer group, which pain points make the target customers most annoyed, and determine the consumption scenarios where customers would buy products or services without hesitation.

The principle of this stage is to "only chase one rabbit at a time". If the team cannot focus on just one important target customer, meaning they want to chase too many rabbits at the same time, they will not be able to serve any one customer group well.

The output of this stage should also be uploaded in the Mini Program to light up the icon for this stage. Another advantage of each team uploading their outputs to the Mini Program is that the facilitator can check the progress of each team in real time, and the teams can also check others' progress.

(4)   Stage 4: diverge the solutions

Once the target customers and their demand scenarios have been determined, the team enters the second divergence–convergence process, in which they discuss and identify solutions to solve the customer pain points in consumption scenarios. The objective of this divergent stage is to individually identify as many kinds of solutions as possible.

There is a tool, named Crazy Eight, to help team members to generate more ideas. The specific method is to fold a piece of A4 paper in half three times to form eight areas. In these areas, each member fills in as many different solutions as possible according to the target demands determined in the previous stage. It does not necessarily mean that they have to come up with eight solutions, but they should try to find as many as they can. The Crazy Eight tool is used to inspire the outputs of individual brainstorming. In this stage, the more creative the ideas, the better. This stage just involves collecting solutions, and criticisms can be saved for the next stage.

(5)   Stage 5: decide on the solutions

After each team member completes their Crazy Eight, the team has a group discussion. The goal of this stage is to converge on one solution that the team thinks is the best to further develop a prototype in the next stages. An innovation matrix could be used in this stage to help teams select the best solution. The innovation matrix is a decision-making tool with two-dimensional coordinates—the abscissa is the degree of demand, and the ordinate is the technical feasibility.

The specific usage of this tool is as follows: (1) number each solution. The solutions previously generated using the Crazy Eight tool are numbered from 1 to 8 here. (2) Place each solution number into the matrix. After discussing the solutions one-by-one, the solution number is placed in the matrix according to its degree of demand and technical feasibility. In the coordinates of the innovation matrix, those with strong demand are placed on the right, and those that are technically easy to implement are placed at the top. (3) Select the best solution. The solutions in the upper right corner, the first quadrant of the coordinate, in the innovation matrix have strong demand and high technical feasibility and should be the most preferred solutions. If there are no options in the first quadrant, the team discusses choosing a suboptimal solution. The selected solution is the one that will be implemented and explored further.

(6)     Stage 6: produce the prototype

Prototyping starts in this stage based on the solution decided upon in the previous stage. The objective of this stage is to generate various prototype ideas divergently and present them visually. The participants collaborate on producing a prototype that most intuitively presents the form, content, and function of the product. Regarding the activities of the Fire Festival, despite the constraints of time and conditions, there are still many ways to make prototypes, such as drawing pictures, making digital posters, or making a sample with simple materials. There are many free-to-use applications available online such as tools that help users generate logos, posters, websites, and videos very easily in a short time.

(7)     Stage 7: validate the prototype

This is the last stage of the three-time divergence and convergence processes. The goal of this stage is to converge on the best prototype and invite other teams to try the prototype as users and obtain their feedback.

It should be noted that verification is not the same as selling. The teams should not try to sell their products to others, but invite users to experience them and think aloud their feelings and feedback. The principle of this stage is "do not defend, just record". If users raise queries and opinions, the team should not make justifications, but rather truthfully record the queries and suggestions for later discussion. After collecting these user opinions, the participants return to their own group to discuss whether the prototype performs well, and to determine the direction of the next iteration of the product or service.

(8)     Stage 8: give a roadshow speech

In this stage, the CEO leads the team to carry out a one-minute roadshow. A tool called Roadshow Canvas is used to conceive roadshow outlines, and the following six issues that investors are most concerned about are addressed: (1) what is the product? (2) who are the users? (3) what channel do we sell through? (4) how do we price for profit? (5) why should consumers choose our products? (6) why should investors invest in our team?

After the roadshow, the audience votes. At this time, everyone changes their roles from entrepreneurs to investors. Everyone, including students and teachers, can vote for one project except their own. The participants vote for the most valuable projects, rather than the facilitators, to decide the results. The Fire Festival Mini Program is also updated to support online voting. The participants can vote by clicking the Like button in the lower right corner of the project, which makes voting and counting votes very easy and fast. An online commenting feature is also provided where participants can post comments below projects.

(9)     Stage 9: reflect on the whole process

The final stage is to review and reflect on the entire process. Reflection after activities such as playing, empathizing, creating, and experimenting plays an important role in students' absorption of the knowledge and experience they have learned. Babson College also places great emphasis on reflection after practice in its entrepreneurship education system [67].

A framework for reflection called good–bad–learn is provided in this stage. First, team members individually reflect on what went well, what went poorly, and what they have learned. Then, they share their reflection with the team. Finally, each group makes a presentation about their reflection and uploads it into the platform, and the nine stages of the Fire Festival have all been completed.

The participating students are expected to accomplish all of the above nine stages within three hours if the Fire Festival is held as an activity or event at one time. Meanwhile, the Fire Festival can also be divided into two or three parts to be held over several sessions according to the teacher's needs. It can also be integrated into the curriculum of their entrepreneurship courses. The various teaching modes of the artifact are elaborated in the next section.

### 4.1.2. Teaching Modes of the Incubation Artifact

The artifact can be used in different ways for entrepreneurship education in universities. In terms of forms, the Fire Festival process could be:

(1) held as an event at the beginning of entrepreneurship courses. In this scenario, the nine stages of the process are expected to be completed within three hours. It can be held continuously in half a day, or it can be divided into two to three weeks of lessons in the class. The main purpose is to allow students to quickly experience entrepreneurship over three hours at a very low cost.

(2) Repeatedly used. Because of the low cost of hosting, teachers can use the artifact repeatedly over the course of a semester. For example, after a quick experience at the beginning of the course, it can be used again in the middle of the semester. The purpose of the second use is different from the first, which is to allow students to come up with a more thoughtful project idea after learning the theoretical knowledge. Subsequent uses can be more focused on the substance of the entrepreneurial project.

(3) Integrated into the courses of other disciplines. The artifact is not only used in entrepreneurship courses, but can also be widely used in courses of other disciplines. Through integration with other disciplines, the artifact could play roles in stimulating students' innovative thinking and provide them with methods for innovation. For example, incorporating the artifact into a robot design course allows students to design robots that are more human-centered, rather than just innovations based on technological progress.

(4) Held as a competition on campus. For teaching-related administrators rather than teachers, this artifact can also be used to organize Fire Festival competitions across the school to stimulate students' entrepreneurial enthusiasm and enhance the atmosphere of innovation and entrepreneurship.

In terms of conditions, the artifact can be utilized online and offline. The software of the artifact was first designed for face-to-face activities; however, due to the global COVID-19 pandemic, the demand for online education has surged. Therefore, the functional iteration of the software over the past two years has focused on supporting the online version of Fire Festival. In response to teachers' and students' feedback, we have developed new functions such as online voting for proposed ideas and online team formation, and now the software can support Fire Festival online.

### 4.1.3. Extended Triple Diamond Model for Entrepreneurship Education

Based on the theory of design thinking and the Double Diamond model, the artifact in this study adopts the process of three "diamonds", i.e., three processes of divergence and convergence. After evaluating the artifact both theoretically and empirically, this research proposes an Extended Triple Diamond model. As shown in the upper part of Figure 5, by keeping three processes of divergence and convergence in the middle, inputs and outputs are added in the extended model.

According to the organizational knowledge creation theory [63], the prime movers in knowledge creation are individuals. Therefore, the input in the model is individuals with tacit knowledge. Before the incubation process begins, a self-organizing team should be built to provide an interaction field bringing an individual's tacit knowledge into a social context where continuous dialogue among team members and the conversion between tacit and explicit knowledge occurs. The output in the model is a prototype that embodies new knowledge that students create through project-based learning, which could be further developed into an entrepreneurship project.

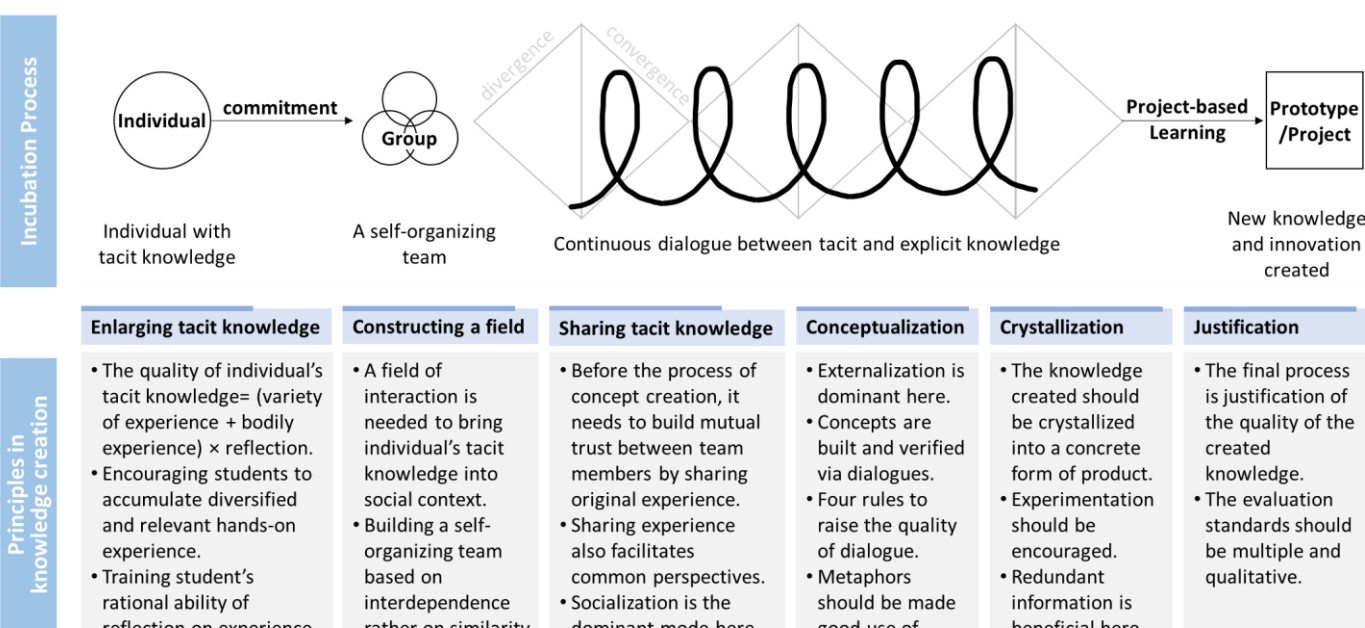

**Figure 5.** Extended Triple Diamond model with design principles in knowledge creation.

Furthermore, along with the process of organizational knowledge creation [63], design principles in the process are provided, as listed in the lower part of Figure 5. The design principles provided here can be beneficial for teacher and student users.

(1) Enlarging tacit knowledge. By adding the inputs into the model, individuals and their tacit knowledge are emphasized. Before the incubation activities, university teachers could help to increase students' tacit knowledge by encouraging them to accumulate diversified and relevant hands-on experience and training their rational ability to reflect on the experience.

(2) Constructing a field. A field should be constructed to allow the dialogue to happen. The field for interaction in this artifact is a self-organizing team. A self-organizing team rather than a team assigned from the top-down gives members enough autonomy to interact with others and enough flexibility to accommodate the complexity and diversity in the context during the creation of new knowledge. The team should be built based on interdependence instead of similarity; therefore, it is necessary to provide a two-way choice between the team leader and the team members in the first stage of team-up.

(3) Sharing tacit knowledge. Before the difficult task of concept creation, mutual trust is needed between team members. A good way to build mutual trust is to share each other's original experience. Therefore, the sharing of participants' original experience should be encouraged throughout the nine stages of the Fire Festival, especially in the beginning, to cultivate mutual trust between team members. The tacit knowledge of individual members is brought into the field by communicating and co-experience, and is converted under the mode of Socialization.

(4) Conceptualization. The mode of Externalization is dominant in the process of conceptualization, involving the conversion from tacit knowledge to explicit knowledge, which is the most challenging and difficult part of the incubation. Concepts are built and articulated through continuous meaningful dialogue among team members. Assumptions can also be tested and verified in the dialogue. There are four field rules [63] to raise the quality of the dialogue: (1) the dialogue should be inclusive and multifaceted rather than conclusive and single-faceted; (2) team members in the dialogue should be able to express their ideas freely and honestly without any social pressure; (3) criticism should be constructive instead of merely blaming; and (4) the dialogue should have temporal continuity to allow the affirmation, negation,

and other volatile dimensions of dialogue to be synthesized to form new knowledge. Another good principle in the mode of Externalization is to make good use of metaphors, which is conducive to the conversion process from tacit knowledge to explicit knowledge.

(5) Crystallization. The knowledge created through the dialogue of team members should be crystallized into some concrete form of a prototype or product. Internalization is the critical knowledge conversion mode here. Action and experimentation should be encouraged in this process to test the concept and the prototype created. In the third diamond of the artifact in this study, the student teams are required to develop a prototype of their ideas and collect "customer" feedback from other groups. They can then refine their prototype according to the testing and feedback. In addition, redundant information among team members is beneficial here. Therefore, members of the team are better off having overlapping responsibilities. The overlap and interdependence among members allow for interactive inquiry, thereby facilitating the process of realizing concepts.

(6) Justification. The final process is the justification of the quality of the created knowledge. The last two stages of the Fire Festival are roadshows and reflection, where student teams can justify their prototypes that have received investors' votes and comments from other teams. The evaluation standards may be multiple and qualitative.

### 4.2. Evaluating: Empirically and Theoretically

This section is an evaluation of the educational artifact. First, the qualitative data from the in-depth interviews with 15 university teachers and students are analyzed and coded into different categories by thematization. Then, the incubation practices of the artifact are evaluated both theoretically and empirically. Comments and suggestions for improvement are given after the evaluation.

#### 4.2.1. Feedback from Teachers and Students

The interview transcripts were analyzed using NVivo 12 Plus. First, all of the interview transcripts were imported into NVivo as a case and coded line-by-line. Themes emerged during the coding process based on the data rather than being predetermined. Then, the codes and themes were individually double-checked separately at the file and code level and fine-tuned when necessary. Finally, matrix coding queries were conducted to compare the different perspectives between teachers and students on specific topics.

(1) Usage scenarios of the artifact

The first feedback to report is the different usage scenarios of the artifact, which means how the users utilize the artifact in practice. As shown in Table 5, according to the feedback, most of the teachers choose to integrate the artifact with their entrepreneurship curriculum and usually use it as the practice part of the course. Some teachers even use this practice to test students' entrepreneurship learning effectiveness as part of their course assessment. Another important scenario is to organize entrepreneurial events or competitions in their school. Furthermore, since the popularity of entrepreneurial contests such as the "Internet+" has increased, teachers usually use the artifact as a convenient tool to guide and incubate student projects for those competitions. Some teachers even think that this artifact can be used to organize ideas for a wider range of project types.

(2) Difficulties and obstacles

The difficulties and obstacles teachers and students encountered when using the artifact were one of the focuses of the evaluation. The feedback was classified into two dimensions. As shown in Table 6, the first is the process of the artifact, from which we can determine which steps they have problems with and what the obstacles are. The second dimension is general limits and constraints, which are difficult to classify into a specific step in the process.

**Table 5.** Different usage scenarios of the artifact.

| Usage Scenarios | Files (8 in Total) | Reference Points | Examples of the Original Data |
|---|---|---|---|
| Integrate with entrepreneurship curriculum | 6 | 11 | T01: I have been using the Fire Festival in class since April last year to inspire students to put forward and sort out their own ideas in class with the help of the Fire Festival tools. T03: Last year we learned about the Fire Festival, and we thought it was great. Then, we figured out how to integrate it into the curriculum. So, in the second semester of last year, we (four teachers) started to use it in 12 class hours as micro-practice. |
| Organize events or competitions | 6 | 7 | S02A: I first came into contact with the Fire Festival because Mrs. Zou held a Fire Festival activity in our school. T02: In terms of entrepreneurship practice, we have regular activities every month, and we hold the Fire Festival monthly. T03: In addition, we also have a student entrepreneurship club in the Maker Space, and we let the students of the club hold the Fire Festival activities two times this semester, which were organized by the students themselves. |
| Guide and incubate projects | 3 | 7 | T03: We set up this micro-practice to allow them to produce several good projects to participate in contests like "Internet+". T04: Many students came to me and said that they have some ideas they want to develop, which is more or less a naive startup project, and this Fire Festival is a good tool to guide them through the processes of it to organize the project. |
| For other activities | 2 | 3 | T01: I even built the Virtual Joint Teaching and Research Office into the Fire Festival as a project, and I used this tool to organize my thoughts. I also have an idea that I can use the tool Fire Festival to organize any kind of project. |
| Total | 8 | 28 | |

In terms of the process, the difficulties are more concentrated in the earlier steps, namely, generating "crazy and stupid ideas" and the first divergent–convergent process of understanding and defining the demands of target customers. The most prominent difficulties in generating "crazy and stupid ideas" are that, on the one hand, it is difficult for teachers to guide, since the ideas that come out are often extreme. On the other hand, it is difficult for students to realize these ideas in the form of a prototype later on. In the first diamond, due to a lack of related concepts and knowledge, students usually have difficulty developing a customer persona and focusing on a specific target segment. Furthermore, some students even insist on not converging to a specific customer group because they believe that their products, such as restorable microbial building materials, have a wide range of application scenarios and user groups. Some other students have a misunderstanding that the more application scenarios of their products, the better; however, often the products that anyone can use are used by no one. These obstacles and misunderstandings reflect the impact of students' insufficient knowledge reserve regarding relevant entrepreneurial concepts.

In response to the above difficulties, the teachers made personalized modifications to the artifact in their practice, as shown in Table 7. Most of the teachers changed the rules due to the difficulty of guiding "crazy and stupid ideas", and others due to the limit of time. The feedback on the process was positive. Only one teacher changed the order of the grouping and raising ideas because of the demands of his actual class, and another teacher improved tools, which were updated in the artifact already. All of the teachers interviewed added more or less support work according to the actual situation of their class, such as sending supporting documents to students in advance to familiarize themselves with the process, explaining the relevant entrepreneurial concepts in advance, or providing some popular entrepreneurial fields for reference.

**Table 6.** Difficulties and obstacles encountered when using the artifact.

| Classified by | Items | Files (8 in Total) | Reference Points | Examples of the Original Data |
|---|---|---|---|---|
| Process of the artifact | To generate "crazy and stupid ideas" | 2 | 4 | T03: I think the stupidity defined here is the kind that can change an era, but this kind of project is beyond my ability as a teacher to guide. |
| | First Diamond—Demands | 5 | 9 | S02A: The most difficult part is describing a client portrait. Because we did not know the customer group very well, there were many disagreements. T03: Students often have a misunderstanding, that is, the more application scenarios of the product the better. They are not willing to converge to a specific customer group. |
| | Second Diamond—Solutions | 2 | 2 | S03C: It took us a long time to position the product. We didn't know at the time whether to locate it as a virtual online or offline. |
| | Third Diamond—Prototypes | 3 | 9 | S03B: I think the prototype verification is difficult. Because our protective glasses product was a kind of object, you had to really make a physical prototype, the customer would give feedback about its advantages and disadvantages. However, we could not make a physical prototype at that time, we could only describe it through language. In that case, it was very difficult to verify the prototype. |
| | Roadshow and reflection | 2 | 4 | T04: Another one, that doesn't go very well is the road show. The students were already a little distracted by the fact that they began to give comments and likes to projects on the software at the beginning of the roadshow. |
| | Subtotal | 7 | 28 | |
| General limits and constraints | Unfamiliar with tools and processes | 2 | 2 | S02A: Because people are not familiar with the process, it is difficult to organize and guide them to go through the three diamonds. T02: I think our students also have some understanding problems when using the tools. |
| | Students lack entrepreneurship related concepts | 2 | 3 | T03: The students are green hands, they do not understand the business model, persona, and related concepts. So, when it comes to diverging and converging customer demands in the Fire Festival, they have problems. |
| | Barriers to online collaboration | 3 | 3 | S01C: Because of the epidemic, the event was held online, and there was still a lot of inconvenience in communication. |
| | Very tight schedule | 1 | 2 | T04: There was not enough time for each group to reflect separately, because at that time, first, the class would be over, and second, they had been sitting for about three hours and had little patience to listen to other teams' presentations. |
| | Subtotal | 6 | 10 | |
| | Total | 7 | 38 | |

**Table 7.** Modifications made to the artifact by teachers.

| Modifications | Files (8 in Total) | Reference Points |
|---|---|---|
| Change the rules | 4 | 13 |
| Change the process | 1 | 1 |
| Add support work | 5 | 7 |
| Improve tools | 1 | 2 |
| Total | 5 | 23 |

(3) Significance and value of the artifact

The teacher and student users' feedback on the significance and value of the artifact was coded into four aspects, as shown in Table 8.

**Table 8.** Significance and value of the artifact.

| Aspects | Items | Files (8 in Total) | Reference Points |
|---|---|---|---|
| For teachers | Helpful for teaching | 3 | 5 |
| | Digital education | 1 | 6 |
| | Project guidance tool | 3 | 4 |
| | Influence and motivate others | 3 | 4 |
| | Achievability | 2 | 4 |
| | Subtotal | 4 | 23 |
| For students | Inspire and express ideas | 5 | 19 |
| | Entrepreneurial experience and passion | 3 | 14 |
| | Improve ability and mindset | 5 | 22 |
| | Boost self-confidence | 4 | 9 |
| | Collaboration | 2 | 7 |
| | Subtotal | 6 | 71 |
| For projects | Generate ideas | 2 | 6 |
| | Organize and incubate projects | 6 | 15 |
| | Digital platform | 3 | 3 |
| | Subtotal | 7 | 24 |
| For university | Climate for entrepreneurship | 1 | 1 |
| | Entrepreneurial ecosystem | 2 | 3 |
| | Education and cultivation | 2 | 3 |
| | Subtotal | 4 | 7 |
| | Total | 8 | 125 |

First, for teachers, the most salient value of the artifact is its aid to teaching. One of the teachers specifically mentioned its significance for education digitization by providing an interesting and effective digital platform for entrepreneurship course teaching. It could also be used as a tool to guide entrepreneurial projects for contests. Some teachers even influenced other colleagues when using the artifact. The teachers felt a strong sense of achievement when they saw the students' outcomes through the artifact.

Second, the students emphasized its value in terms of inspiring and expressing ideas that may seem crazy and stupid to others. Expressing themselves freely and interacting with others not only improved the students' comprehensive abilities, but also boosted their self-confidence, especially among students from vocational colleges. At the same time, many students mentioned that this activity was their first experience of entrepreneurship, which aroused their interest and enthusiasm for entrepreneurship.

Third, for projects, the artifact helps generate entrepreneurial ideas on the spot from nothing but the tacit knowledge of students. They both mentioned that the artifact is particularly suitable for organizing projects in the early stage. Its value is also reflected in the fact that, as a digital platform, all project and interaction data are stored on the platform, and as digital assets, they can be very conveniently displayed and reviewed.

Lastly, for universities, the artifact is conducive to a climate of entrepreneurship since it spreads quickly among students. It works as an essential supplement in a bottom-up approach in the entrepreneurial ecosystem of universities by educating students and incubating both talent and projects.

(4) Different views between teachers and students

The reasons teachers and students are attracted to the artifact and their level of agreement on generating "crazy and stupid" ideas are compared in this section, and some

interesting differences are reported. The reasons why teachers and students were attracted to the artifact are classified into four categories, as shown in Table 9.

**Table 9.** Reasons for teachers and students being attracted to the artifact.

| Reasons | References of | | Examples of the Original Data |
|---|---|---|---|
| | **Teachers** | **Students** | |
| Low entry barrier | 12 | 1 | T01: I feel like the Fire Festival has no threshold, which is what I like best. All ordinary people, even if you don't have an idea, can participate and we generate ideas on the spot. T04: The process is relatively simple and easy to use, which is one of the things I find attractive. |
| Fun and novelty | 5 | 7 | S01C: When I went to participate, I was holding a kind of curiosity, and I was eager to try it. S03A: What attracted me to the Fire Festival in the first place was the format of the activity, which is kind of like a game. T04: The biggest value I just said is that it can be used as practice in the form of a game in our class to make the theory more interesting. |
| Tools and process | 6 | 1 | T01: I like the design thinking process of it. There are three divergent and three convergent links that I really like. T05: Something must be able to be broken down into steps, each step is doable, and it can be done. So when I met the Fire Festival with three divergent convergence, I thought this pattern was very good. S01C: Yeah, it's a little bit more in line with my personality, and I feel like this is something I want to try to do. |
| Meet their needs | 4 | 2 | T05: Later, we want to change the technology-based innovation (innovation 1.0) to the robot innovation based on social needs (innovation 2.0), then the original method has to be changed, and we have to use the Fire Festival. Its several times of divergent and convergent exactly meet the needs of my course. |

In order to better distinguish the differences between teachers and students, a radar map of the reasons why they are attracted to the artifact was devised and is shown in Figure 6. By doing so, we could clearly see the different distribution between teachers and students. Four out of five interviewed teachers emphasized the low entry barrier of the artifact. Moreover, from the point of view of the facilitator of the activity, it is also very easy to organize, while from the view of the students, the most salient reason why they wanted to participate from the very beginning was that it is fun and novel, and they very much enjoyed the whole process of playing. All three groups of students emphasized this dimension, and two teachers valued this. However, there was a slight difference between the fun and novelty they mentioned: since the teachers were organizers rather than participants, their emphasis was more on making their courses and teaching interesting.

The most controversial part is the starting point, generating "crazy and stupid ideas". In the interviews, the researcher set a separate question for this issue to obtain more detailed feedback. The coded feedback was classified into three types: strongly agree, somewhat agree, and disagree, according to their level of agreement on this practice. In Table 10, both the number of reference points and files at different levels of agreement are listed and the examples of the original data are given. In the category of "somewhat agree", there are two situations. Three files, with seven reference points, indicated that the respondents felt that the ideas generated under the guidance of crazy and stupid principles were hard to carry out. Five files, with the other seven reference points, were in agreement with the conditions. These respondents admitted that the principle itself was good, but it was beyond their ability to guide students on the right track.

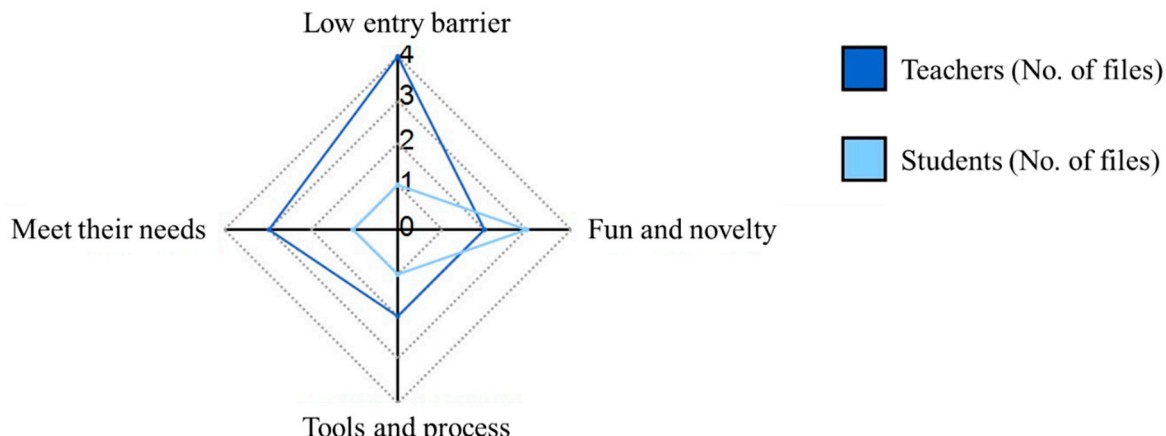

**Figure 6.** Radar map of reasons teachers and students are attracted to the artifact.

**Table 10.** Level of agreement on generating "crazy and stupid ideas".

| Level of Agreement | References | | Files of | | Examples of the Original Data |
|---|---|---|---|---|---|
| | T | S | T | S | |
| Strongly agree | 4 | 5 | 1 | 3 | S03D: I think it's very good because there are very few opportunities to express your ideas to your peers, with your own ambition or passion. T02: I think this is the best part since every student is the protagonist. When you go to the later part, not every student is the protagonist, maybe the person in charge is the protagonist. |
| Somewhat agree | 10 | 4 | 3 | 2 | S03A: Although this starting point is very good but at the stage of product prototype, I found my idea was a unrealistic and hard to carry it out. T03: The crazy ideas may seem stupid but they may change the world. Unfortunately, it's beyond ordinary teachers' ability to guide. |
| Disagree | 1 | 0 | 1 | 0 | T01: Once, I wrote news about the Fire Festival, in which I used the term "crazy and stupid ideas". However, the leaders of our publicity department asked me to change the term because they didn't understand. |

From the heat map shown in Figure 7, we can see that the students' opinions focus on "strongly agree" and the teachers' attitudes are mainly partial agreement. In particular, it should be noted that the heat map here is drawn according to the number of reference points instead of the number of cases. The reason is that some participants referred to the attitude of other stakeholders, whose attitudes differed to theirs. Therefore, it is not suitable to use the number of cases here as it will result in a situation whereby the same respondent both strongly agrees and partially agrees.

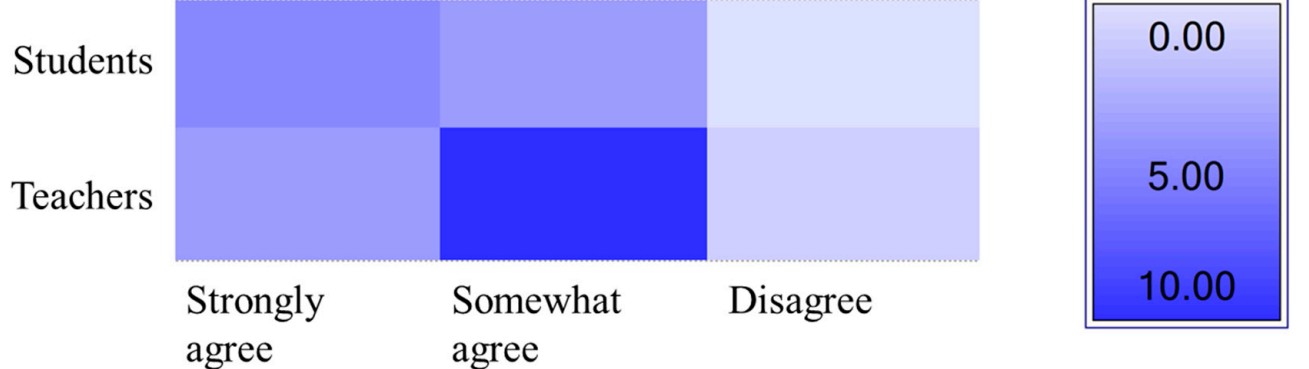

**Figure 7.** Heat map of the level of agreement on "crazy and stupid ideas".

4.2.2. Evaluation through the Lens of SECI

In this section, the artifact is evaluated through the lens of the SECI model of organizational knowledge creation theory, combined with the coded themes of the empirical study. The evaluation of the artifact is summarized in Table 11. According to Nonaka's [63] dynamic theory of organizational knowledge creation, there are four modes of knowledge conversion:

(1) Socialization, from tacit knowledge to tacit knowledge, in which the trigger is shared experience between team members. Positive feedback was given by teachers and students on the roles the artifact played in constructing a community and creating an atmosphere of innovation and entrepreneurship.

(2) Externalization, where the knowledge conversion is from tacit knowledge to explicit knowledge, which is crucial for the artifact. This marked with an asterisk in the table. The trigger in this mode of conversion is a continuous meaningful dialogue between team members. The three divergence and convergence processes in the artifact provided students with successive rounds of dialogue. However, the use of metaphors was not enough for either students or teachers.

(3) Combination, which is the mode of knowledge conversion from explicit knowledge to explicit knowledge. The online platform and tools of the artifact played an important role in the exchange and Combination of explicit knowledge between team members. At the same time, the features of these online platforms and tools need to be continually improved to alleviate the challenges associated with conducting activities online.

(4) Internalization, which is the conversion from explicit knowledge to tacit knowledge in the SECI model. Action and experimentation are highly related to this process. The easy-to-start and cost-effective features of the software were conducive to students' trial-and-error. Comments and suggestions after being evaluated theoretically and empirically are given in the last column of Table 11.

**Table 11.** Evaluation of the artifact through the lens of knowledge conversion modes.

| Knowledge Conversion | Mode | Triggers of Each Mode | Feedback on the Use of the Artifact | Comments and Suggestions after Evaluation |
|---|---|---|---|---|
| Tacit → Tacit knowledge | Socialization | The Socialization mode usually starts with the building of a team or field of interaction. People share each other's thinking processes through shared experiences. | ✓ Students gained experience and growth in entrepreneurship. ✓ Conducive to creating an atmosphere of innovation and entrepreneurship. ✓ Established a community of entrepreneurship. | In holding a Fire Festival through the same school or major, students usually have shared experiences. However, if the activities are held across schools, the shared experience between team members of a group should be emphasized. |
| Tacit → Explicit knowledge | Externalization | The Externalization mode is triggered by successive rounds of meaningful dialogue. Metaphor plays an important role in the process of dialogue. | ✓ The routine of three divergence and convergence processes is good, making entrepreneurship achievable. ✓ The underlying logic of the activity uses the theory and principles of design thinking. ✓ The comments from judges of Fire Festival focus on guiding projects, which is helpful dialogue between students and experts/teachers. | The three divergence and convergence processes in the Fire Festival provides students with a successive round of meaningful dialogue, although in an intensive fashion. The use of metaphor is not specifically emphasized in the current guide, which can be improved in the next iteration. |
| Explicit → Explicit knowledge | Combination | The Combination mode is facilitated by coordination between team members and the documentation of existing knowledge. | ✓ A good data platform to record creative ideas. The digital achievements are deposited on the platform, which is convenient to display at any time. ✗ Hosting activities online brought challenges to team collaboration. | The online platform and tools of Fire Festival play a crucial role in exchanging and combining explicit knowledge. Features of online platforms and tools need to be continually improved to better support online collaboration. |
| Explicit → Tacit knowledge | Internalization | The Internalization mode is triggered by experimentation, an iterative process of trial and error. | ✓ Easy to start and very cost-effective, therefore can be used over and over again. ✗ Inadequate understanding of some of the nine stages of the Fire Festival. | Provide teachers and students with a workbook and training to help them internalize the explicit knowledge of Fire Festival's process and tools into their tacit knowledge. |

Legend: ✓ means positive feedback. ✗ means negative feedback.

## 5. Discussion

Artifacts are designed for certain purposes. In this section, the educational artifact is first compared with similar models to better understand its design intentions. Second, the implications of the study are given. Lastly, the limitations and future research are discussed.

### 5.1. Comparing the Extended Triple Diamond with Similar Research

Compared with Double Diamond, the third stage of design thinking, "ideate", is listed as a separate divergence–convergence process in the artifact of this research to ideate solutions for the defined problems, making the process a triple diamond. The Extended Triple Diamond method makes the teaching rhythm smoother. In this model, inputs are added to emphasize individuals' tacit knowledge, and outputs are added to crystallize the new knowledge created in the three diamonds into some concrete forms of explicit knowledge. The "extended" part of this model visualizes the process of knowledge creation from tacit to explicit.

Although the six core stages of the triple diamond may look like the steps of Google Design Sprint, the two are designed for totally different scenarios. The artifact of this study is tailor-made to meet the demands of entrepreneurship education in universities. It can be conducted within three hours rather than the five days it takes to complete Google Design Sprint, and can be merged into the curriculum of entrepreneurship courses. Besides the process, we also provide supporting software to scaffold both teacher and student users. Because of the easy-access software, the barriers to using the artifact are considerably lowered, contributing towards meeting the sustainable development goals of entrepreneurship education.

### 5.2. Theoretical and Practical Implications

This design science research contributes to the body of design knowledge in three ways: an empirical design object, a theoretical design object, and empirical design evaluation, according to the design knowledge contribution framework [68]. First, the educational artifact created in this study is an empirical design object contribution. Describing the tasks and rules of each stage in the incubation process provides an empirical understanding of how to use the artifact. Second, this research proposed an Extended Triple Diamond model for incubating entrepreneurship from tacit knowledge, which is the theoretical design object contribution. Besides this, the design principles were given to raise the quality of the knowledge in the knowledge creation process. Third, this research empirically evaluated the artifact, which is the empirical design evaluation contribution. The feedback from participant teachers and students helped with iterating the artifact.

Regarding the implications for practice, the artifact, with its gamified educational software designed in this study, has been widely used across universities and colleges in China with more than 80,000 student users and over 1000 teacher users. Empowered by the easy-access software, users can generate creative ideas and further develop them into prototypes or projects at a meager cost and with low entry barriers. By doing so, the inherent contradictions and conflicts between the personalization of innovation and entrepreneurship (usually at a high cost) and large-scale development (conducted in a sustainable manner in China) can be alleviated. In this sense, the educational artifact of this study responds to the challenges of the sustainable development goal of innovation and entrepreneurship in China, contributing to the SDG on education.

### 5.3. Limitations and Future Research

This design science research adopted a conventional article structure to organize and present the content. According to the latest theoretical guide for the design science research of entrepreneurship outlined by Dimov [68], a creating–evaluating–theorizing–justifying structure could be used to better show the cyclical nature of the design science research; the

theorizing and justifying parts should also be strengthened. Quantitative studies targeting a wider range of respondents could be conducted in the future using specific criteria.

## 6. Conclusions

To meet the challenges associated with the sustainable development of entrepreneurship education in China, this study adopted design science methodology to create an educational artifact designed to incubate entrepreneurship from the tacit knowledge of university students. Besides the process and tools of the artifact, an Extended Triple Diamond model was proposed with the incubating principles included in the knowledge creation process. An evaluation was conducted after the creation of the artifact. University teacher and student users were invited to take part in in-depth interviews in order to obtain their feedback. Through the evaluation, the process and tools of the Fire Festival have been iterated, and the functions of voting for creative ideas and the formation of teams have been added to better support online teaching.

**Funding:** This research was funded by Chongqing Municipal Education Commission, grant number 234074, and Sichuan International Studies University, grant number JY2380269.

**Institutional Review Board Statement:** The study was approved by the Ethics Committee of ELM Graduate School, HELP University (date of approval: 10 June 2022).

**Informed Consent Statement:** Informed consent was obtained from all subjects involved in the study.

**Data Availability Statement:** The interview data presented in this study are unavailable due to privacy and ethical restrictions that protect the teachers and students who took part in this research.

**Acknowledgments:** The author wishes to thank Bernard Lew Shian Loong and Bailing Zheng for their continued guidance and supervision; Wendy Liow for her many good suggestions during the topic selection; Nicholas Lum for organizing seminars; and Chung Tin Fah, Edmund Oh, Mohan Raj, Thang Siew Ming, Yap Kim Len, and Paolo Casadio et al. for their suggestions in the seminars. The author would also like to thank the initiators of EECN, Yankong Zhu and Jiansong Yin, and all of the working partners in EECN for providing opportunities for practitioners of entrepreneurship education. Lastly, the author wants to thank the three reviewers involved in the peer review process for their great suggestions on revisions.

**Conflicts of Interest:** The author declares no conflict of interest. The funders had no role in the design of the study; in the collection, analyses, or interpretation of data; in the writing of the manuscript; or in the decision to publish the results.

## Appendix A

*Appendix A.1. In-Depth Interview Guide for Teachers (Translated from Chinese)*

Type of research activity: In-depth Interview with Teachers
Participant identification number:
University:
Gender (Male/Female):
Interview date:

1. Please describe your position in your organization (name of institution). How long have you been in this position?
2. What work are you mainly engaged in related to innovation and entrepreneurship (incubation)?
3. Does your university have any entrepreneurship and innovation institutions (such as school of innovation and entrepreneurship, innovation and entrepreneurship centers, etc.) and entrepreneurship incubators (such as incubation parks, industrial parks, makerspaces)?
4. When did you start using Fire Festival? How do you use it (in conjunction with courses or organizing competitions)?
5. What attracted you to the Fire Festival? How is it different from other official competitions such as "Internet+"?

6. How did the "Internet+" project of your university come into being (the source of the project), and does the university have any relevant incubation measures?
7. In which of the nine parts of the Fire Festival do you often encounter challenges (difficulties)? If any, please elaborate.
8. How do you guide students to "crazy and stupid ideas" when organizing a Fire Festival?
   (a) Do you always start with "crazy and stupid ideas" when facilitating a Fire Festival? Do you have any other practices?
   (b) What do you think of the idea that the starting point of the Fire Festival is always a "crazy and stupid ideas"?
9. What are your suggestions for the nine stages and online support tools of the Fire Festival?
10. What do you think is the significance and value of Fire Festival?
11. That is all the questions I have prepared for today. Do you have any questions about this interview, or do you have anything else to add to your answers above?

*Appendix A.2. Focus Group Interview Guide for Students (Translated from Chinese)*

Type of research activity: Focus Group Discussion with Students
University:
Interview date:
Participant identification number:
Gender:
Grade and major:
The Fire Festival project and the role in the project:

1. What is your major and grade? Please explain your role in the project.
2. What innovation and entrepreneurship related courses and activities have you participated in?
3. Does your university have any entrepreneurship and innovation institutions and entrepreneurship incubators? Which ones have you been exposed to?
4. When did you start using Fire Festival? How do you use it (in conjunction with courses or organizing competitions)?
5. Have you participated in competitions such as "Internet+"? How did the "Internet+" project of your university come into being (the source of the project), and does the university have any relevant incubation measures?
6. What attracted you to the Fire Festival? How is it different from other official competitions such as "Internet+"?
7. In which of the nine parts of the Fire Festival do you often encounter challenges (difficulties)? If any, please elaborate.
8. How did the facilitator guide students to "crazy and stupid ideas" when organizing a Fire Festival? What do you think of the idea that the starting point of the Fire Festival is always "crazy and stupid ideas"?
9. What are your suggestions for the nine stages and online support tools of the Fire Festival?
10. What do you think is the significance and value of Fire Festival? What was your biggest takeaway from the Fire Festival?
11. That is all the questions I have prepared for today. Do you have any questions about this interview, or do you have anything else to add to your answers above?

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
