# Peer review of "An Entrepreneurship Incubation Process Model and Gamified Educational Software Designed for Sustainable Education"

_sustainability, doi:10.3390/su151914646_

Round 1

Reviewer 1 Report

       i.          Please list the total number of interviewees on line 17.

      ii.          Please start the first few lines of the introductory section with a more generic explanation of the subject without any citations.

     iii.          "In this chapter, the science of the artificial was first introduced..," is written in line 77. Please change the word "the chapter" to a more appropriate word as this is a journal article.

    iv.          Please identify the statement's sources in Table 1

     v.          The part on literature reviews has been skillfully written and extended. However, you fall short in connecting this section to the viewpoint and issues of the entrepreneurship course intended for students and teachers in China. Please do better

    vi.          The Research Methodology section is very beautifully written. Please clarify what Tencent Meeting and Feishu Online Meeting in Table 3 mean, as well as why total word count is included in this table.

   vii.          In general, the writers' writing is excellent, especially in the section on data analysis and discussion. This article may serve as a resource for entrepreneurial instructors all over the world and is useful for understanding or improving the teaching and learning process in the topic. Congratulations!

Minor English editing is required.

Author Response

Dear Reviewer,

For the point-by-point response to your comments please see the attachment.

Thank you.

Reviewer 2 Report

The authors should provide a more detailed explanation of the rationale for selecting Campus VC as the case study. Specifically, how does Campus VC's role in the Quadruple Helix model make it more suitable for the study?

The exact response rates for student interviews and any potential implications should be discussed.

The authors mention that the data analysis was conducted in Chinese and later translated into English. They should elaborate on the process of translation, as to how the translation did not introduce bias or misinterpretation of the data.

The authors should include some discussion about the validity risks and how they were mitigated. For example, were there any potential biases in the data collection process, and how were these addressed?

The authors state that the feedback from teachers and students serves as an evaluation of the artifact. They should include if there are specific criteria or metrics used for this evaluation.

The authors should provide a brief description of Figure 3 to aid in understanding for readers who may not be familiar with it.

The authors should list how many teachers and students were interviewed in total.

The authors should provide a more detailed rationale for choosing the design science methodology approach and how it compares to other possible methodologies.

The authors mention the structure of "Creating-Evaluating" in the research but do not explain how this structure is relevant to the study's objectives. There is a lack of clarity in connecting this framework to the research's purpose.

In summary, there is a lack of crucial details, explanations, and transparency in several areas, including methodology selection, structure, data collection, and potential biases. The authors should address these issues.

The text contains multiple typographical errors and formatting inconsistencies, which take away from the professionalism of the research paper.

Author Response

(The authors gave the same response as above.)

Reviewer 3 Report

This is an interesting research with good research methodology and yet it is far too long. The authors are suggested to separate the contents into two papers, for example, one is about the design of the software whilst another one is to evaluate the usefulness of the educational software for supporting entrepreneurship. It seems the larger part of the paper is related to evaluation of the Fire Festival so the authors might like to focus on this aspect in this paper. As a result, there is a great need to re-write the Abstract, Literature Review, Methodology etc.. to reflect the changes. 

Section 3.3 was clearly presented and it was good to present the facts in tables. However, it is unclear which interviewee questions are related to which research questions. Was there any literature underpinning the research questions? Was there a pilot study prior to conducting the interviews. In fact, it is unusual to bundle different aspects into one research question. You may like to propose broader questions and then delineate them into finer research questions.

Section 4.1 is totally unrelated to Data Analysis and Results. Section 4.2 was clearly presented and yet it was unclear how the findings were related to the research questions. The authors may like to re-write the research questions according to the classifications and /or items.The Discussion Section should be about the findings especially to compare and contrast with similar research rather than on research significance and limitations. I look forward to reading the revised version.

Some sentences were difficult to understand.

Author Response

(The authors gave the same response as above.)

Round 2

Reviewer 2 Report

The previous concerns have been addressed by the revisions made to the manuscript.

Looking forward to seeing your research shared with the scientific community.

Reviewer 3 Report

The responses by the authors are acceptable and the revised manuscript is much clearer and organized. Congratulations.